# A Disposable Pneumatic Microgripper for Cell Manipulation with Image-Based Force Sensing

**DOI:** 10.3390/mi10100707

**Published:** 2019-10-18

**Authors:** Benjamin Gursky, Sebastian Bütefisch, Monika Leester-Schädel, Kangqi Li, Barbara Matheis, Andreas Dietzel

**Affiliations:** 1Institute of Microtechnology, Technische Universität Braunschweig, 38124 Braunschweig, Germanya.dietzel@tu-braunschweig.de (A.D.); 2Physikalisch-Technische Bundesanstalt, 38116 Braunschweig, Germany

**Keywords:** microgripper, pneumatic actuation, image-based force sensor, SU-8, patch clamp technique

## Abstract

A new design for a single-use disposable pneumatic microgripper is presented in this paper. It enables very cost-effective batch microfabrication in SU-8 with a single lithography mask by shifting manufacturing complexity into reusable components. An optically readable force sensor with potential to be used in a feedback loop has been integrated in order to enable gripping with a controlled force. The sensors are first examined separately from the gripper and exhibit good linearity. The gripper function utilizes the disposable gripper element together with a reusable gripper fixture. During experiments, the pneumatically actuated microgripper can vary the gripping force within a range of a few mN (up to 5.7 mN was observed). This microgripper is planned to be used in a liquid environment for gripping larger aggregates of cells in combination with the patch clamp technique. This approach will allow Langerhans islets suspended in an electrolyte solution to be grasped and held during electrophysiological measurements without cell damage.

## 1. Introduction

Several different grippers for microscopic objects have already been developed for commercial use. For example, the company FemtoTools offers grippers of the electrostatically driven FT-G series [1] which is presented scientifically by Beyeler [2]. SmarAct GmbH offers the piezo-driven gripper SG-06 [3]. In recent scientific works, other grippers with electrostatic [4] or piezomechanical [5] actuator principles have been presented. Hoxhold [6] and Garcés-Schröder et al. [7] present microgrippers with actuators made of a shape memory alloy (SMA). In particular, the latter design is intended for the handling of biological samples. A comprehensive list of actuation principles for microgrippers is given by Dochshanov et al. [8], mentioning electrostatic, thermal, electromagnetic, and piezoelectric as the most common examples.

In the context of their application in liquid environments and their use in combination with electrophysiological measurements, the mentioned examples have the disadvantage of being electrically driven. Electromagnetic fields could exert disturbing influences on very weak electrophysiological signals (in the picoampere range) from cells and organoids. Thermomechanical actuator principles, such as the designs described by Hoxhold [6] and Garcés-Schröder et al. [7] (SMA), or by Chronis et al. [9] (thermal expansion) have the additional disadvantage of heat dissipation into their surroundings, which could also influence the samples. If submerged into an electrolyte solution surrounding a sample, some electrically driven actuators can also short circuit. Fluidic actuators, whether pneumatic or hydraulic, are promising because they do not show any heat development worth considering, and do not require electrical components in the vicinity of the sample. A pneumatic microgripper has been developed [10] in which the gripper’s arms perform a rotational movement similar to a pair of tweezers. This could be disadvantageous for a gripped cell cluster as it could be forced out of the tilted gripper jaws. This problematic tilting can be eliminated by moving the gripper jaws in parallelogrammatic elastic hinge structures, as demonstrated by Hoxhold [6]. This pneumatic microgripper design can be seen in Figure 1. The following article focuses on a successor of that design.

To avoid contamination, everything that comes into contact with a biological sample must be either sterilizable or easily replaceable. In the case of single-use objects, economic mass production should be possible. The previous design [6] consists of compliant SU-8 structures processed on a silicon wafer with glass adhered to the top. The pneumatic actuator consists of meandering SU-8 structures that are closed off by the silicon at the bottom and by the glass at the top. This forms a bellows structure, where the meandering part of the SU-8 can slide between the silicon and the glass to generate a displacement and/or a force when applied with either pressure or vacuum. Although it should allow batch production, the process could not be reproduced without adjustments that prevent a full batch process. Originally, the bellows structures were closed on wafer level by adhesion of the glass wafer followed by chip separation. The dicing sludge from the separation, however, can restrict the bellows’ movability if the sludge cannot be removed completely. The sludge has to be flushed out of the actuator element, something that is difficult to achieve when the glass cover is already in place. For a better result, dicing and rinsing with ultrasound has to be carried out first. The glass cover must then be positioned and glued on each microgripper individually.

The measurement of the gripping force is another important aspect in a handling system for cell samples. Forces applied to very small samples of muscle tissue were measured for mechanical characterization using a force sensor featuring compliant, parallelogrammatic silicon structures [11]. The deflection of those structures was determined via piezo resistors that are doped into the silicon. As with the actuators, a non-electric function is preferable for patch clamping. This means that common principles featuring Wheatstone bridges made of piezo resistors presented in various works [6,11,12,13] are not suitable. Gripping arms designed as flexible cantilevers are shown by Alogla et al. [10]. Their deformation can be detected by the reflection of a laser beam and used to determine the momentary gripping force. While this approach keeps electronic components away from the sample, the integration of laser equipment would increase the potential handling system’s overall complexity and, therefore, its cost. This can be avoided as shown below. This article describes a new single-use gripper to be used in a handling system that allows precise positioning and holding of small samples of cell clusters in addition to the patch clamp technique. During patch clamping, the electrical currents through cell membranes are measured in the picoampere range. Common approaches to cell-handling use the suction of a pipette to hold samples, however, prolonged exposure to suction during measurement could affect the sample’s physiology. By contrast, the samples in this new handling system will be held and moved by utilizing a microgripper.

In order to meet these requirements, this article presents a modification of the pneumatic design by Hoxhold [6], which not only allows batch production, but also greatly simplifies the manufacturing process by shifting much of the recurring complexity of the gripper’s manufacturing process into a reusable fixture. In addition, the new design is equipped with a sensor system that enables the determination of the momentary gripping force utilizing the system’s microscope.

The use of such a handling system should enable researchers to work with larger aggregates of cells without damaging them. A conceptual sketch of the gripper in the planned application can be seen in Figure 2.

## 2. New Disposable Microgripper

### 2.1. Micro Manufacturing

The central gripper component is manufactured lithographically in a clean room using a process shown schematically in Figure 3. This process is fully batchable, and requires exactly one lithography mask. It starts with a silicon wafer of 360 µm thickness as a carrier. For better adhesion of the following copper layer, the wafer is sputter-coated with a 10 nm-thick chromium layer. A 2.1 µm thick copper layer is sputtered on top which serves as a sacrificial layer to later separate the structures from the carrier wafer. After oxygen plasma activation, the first layer of the precursor SU-8-50 (Kayaku Advanced Materials, Inc. (formerly MicroChem Corp.), Westborough, MA, USA) is deposited by spin coating. The sample is covered and left for 20 min at room temperature and then for about 1 h at 60 °C for planarization. After planarization, the cover is removed. The temperature is slowly increased to 100 °C and the SU-8 coating is left to dry for 7 h.

Spinning and drying are then repeated, resulting in an SU-8 layer with a total thickness of about 400 µm. The negative photoresist SU-8 is exposed to a dose of about 1500 mJ/cm² of broadband UV light and subjected to a post-exposure bake. To reduce residual stress, the sample is left at room temperature for at least 12 h before development. In the last step, the sacrificial copper layer is alkaline etched for several hours until all structures have detached from the carrier wafer. One of the released disposable gripper components can be seen in Figure 4.

In comparison to the earlier works [6], this process eliminates three of the four required lithography masks (the structuring of the sacrificial layer, ventilation holes in the silicon base, and adhesive for the glass cover). What remains is a single design layer for the SU-8 structures. Since the disposable components are completely detached from the carrier wafer, separating by dicing is no longer necessary. A very simple and cost-effective batch production is established.

### 2.2. Design of the Disposable Gripper Component

Figure 4 shows the disposable component of the microgripper made of SU-8. It exhibits a base of 7.4 mm in width. The total length can be about 10 mm, but depends on the implementation of the force sensors. In the following, a variant with a total length of 9.55 mm and a thickness of about 400 µm will be discussed, which is defined by the thickness of the spin-on liquid SU-8 precursor.

The pneumatic actuator functions like a bellows that consists of both a flexible and a rigid part. The meandering SU-8 structure is flexible enough to allow movement of the rigid pressure barrier in the direction of the compliant transmission mechanism when subjected to a pressure difference across the barrier (i.e., air pressure). The gripper opens and closes when the pressure behind the barrier exceeds or falls below the environmental pressure. The displacement of the rigid barrier is transmitted via the elastic hinge mechanism to the arms of the gripper. The transmission is designed to have exactly one degree of freedom and moves the arms almost symmetrically. Due to the parallelogrammatic design, the gripper jaws always remain parallel to each other. See Figure 5 for a simplified schematic view of the kinematics. The flexible hinge joints have widths of 45 µm and are 500 µm long.

Mechano-optical gripping force sensors are integrated, which consist of two spring elements per gripper jaw and enable a quasi-parallel displacement. The sensors can be read out under a microscope by image-based evaluation. Readout structures that function according to the principle of the vernier scale in a mechanical caliper can be used. The opposing readout marks have distances of 50 µm on one side and 45 µm on the other. This divides the possible deflection range of 100 µm into 20 steps of 5 µm each. The total movement of the microgripper’s jaws consists of the movement induced by the actuator through the transmission augmented by the deflection of the sensors. It is therefore dependent on both the air pressure in the actuator and the current gripping force. The flexibility of the beam springs and, thus, the sensitivity of the force sensors can be adjusted by changing the spring dimensions in the design—extending the springs, for example, increases the sensitivity. The spring length in this work was set to 1200 µm, and the spring width to 45 µm. Finite element analysis using SolidWorks (SolidWorks 2016 × 64 Edition, Dassault Systèmes SolidWorks Corp., Waltham, MA, USA) was carried out to support the design of the force sensors. Figure 6 shows the model used for simulation where the Young’s modulus and the Poisson ratio are set to 5.6 GPa and 0.22 respectively. The interaction of the geometric parameters (beam length, beam width, and SU-8 layer thickness) were investigated. It was found that third-degree polynomials provide acceptable results for approximating the influence of the three parameters on the deflection of the sensor system at a given force. Figure 7 shows an example of the influence of spring beam length and layer thickness in simulated and approximated values.

### 2.3. The Reusable Pneumatic Fixture

The reusable pneumatic fixture for the disposable gripper component was manufactured from aluminum using classical machining methods. The brass tube for a pressure connection and the cylindrical pins for alignment of the partial elements are pressed in.

To allow a pressure difference to build up across the moving barrier in the disposable element, the element is inserted into the pneumatic fixture. The fixture covers the actuator’s open sides with the corresponding aluminum surfaces (Figure 8). The fixture consists of two halves. The lower half has two pins on which the SU-8 part is mounted. This prevents lateral movements and rotation of the disposable component frame. By fastening the upper half, the microgripper is fixed in the vertical direction and the pneumatic actuator element is closed. Secondary mounting braces both receiving halves against each other. In order to optionally reduce or eliminate contact forces between disposable and reusable parts, the mounting uses spacer springs between the halves. These are made of conventional silicone O-rings. Two rings are stacked per centering pin. The rings are exchangeable to allow the use of grippers with other SU-8 thicknesses.

The length of the mounting is 19 mm, and the width is the same as for the disposable gripper. With the gripper inserted, the mounting has a total height of approximately 8.4 mm. It tapers at the tip to about 1.4 mm in height. The pressure supply of the drive in a range of approx. −0.1 to +0.15 MPa is provided by a small tube in the top part of the mounting, which is led diagonally towards the rear and to which a single hose is connected. The compressed air is fed through the top part into the main chamber of the actuator element. The air volume displaced by the actuator element can escape through two exhaust air openings at the top and also through the gaps at the sides of the connection between the actuator and the transmission. There is a gap between the bellows structure and the top/bottom fixture to allow movement of the actuator. This means the full supplied pressure will not build up inside the actuator as air will continuously flow through this gap when pressure—either positive or negative—is applied.

## 3. Experiments

### 3.1. Mechanical Characterization of the Gripping Sensors

In order to evaluate the gripping force sensors and validate their simulation, dummy grippers with immovable transmissions were manufactured. These dummies allow investigations in the test stand without influence on the force sensors due to the compliance of the transmission or the actuator. A commercially available force sensor (KD78, ME-Meßsysteme, Henningsdorf, Germany) is mounted on a linear drive. The sensor can measure forces in a single direction which is parallel to the movement of the linear drive. By moving the axis, a measuring tip attached to the KD78 can be pushed against the parallelogrammatic structures of the gripping force sensor, causing a deflection of the sample in line with the KD78. Meanwhile, the effective force is measured by the KD78 and the deflection distance is shown by how far the linear axis moves.

To determine the zero point of the displacement, the contact point between the measuring pin and the gripping force sensor is determined by gradually closing the distance between the probing tip and the sample until a small force can be measured. Then, they are moved away from each other until the force vanishes. It is always started without force, and the distance to the maximum applied force is recorded twice in both directions. Since the KD78 itself also has a force-dependent deflection of about 0.181 µm/mN, this has to be mathematically compensated for during the evaluation.

The measured values are shown in Table 1. Here, it can be seen that the force sensor systems examined have sensitivities of 3.182–3.428 µm/mN with an average of 3.337 µm/mN. The standard deviation is 0.088 µm/mN. Sensitivities close to 3.3 µm/mN are expected from the simulations and confirmed by the samples. The deviations can be explained by fluctuations in the SU-8 layer thickness, at which fluctuations of 6% or more are possible in extreme cases.

With coefficients of determination R² between 0.9932 and 0.9960, the measured values show good linearity between the effective force and the deflection. Offsets of between −0.063 and −1.187 mN can be measured. These are likely induced by the contact process, which can leave a small gap between the measuring pin and the sample.

### 3.2. Exemplary Microgripper Experiments

In order to demonstrate that the gripping force sensors also function in practical applications, the force sensors of two complete grippers with actuators, transmissions, sensors, and optical readout structures are examined. Since the two gripper jaws of each gripper are symmetrically equipped with force measurement structures, each side (Side 1, Side 2) of the gripper has its own independent measurement.

#### 3.2.1. Force Sensor Calibration

First, the force sensors are calibrated to determine the individual offset and sensitivity of each respective sample. The most crucial aspect is to enable the user to derive the effective gripping force from image recordings of the sensors. For this purpose, the tactile tip on the KD78 force sensor is used again in the setup of the previous measurement. The front part of one of the gripper jaws is touched, and a force is applied to the gripper jaw via the movement of the linear drive. This force is increased until the parallelogrammatic gripping force sensor has reached its maximum deflection. Then, the linear drive returns to the starting position. During the deflection phase, images of the reading structures are recorded and stored at regular intervals. The arms and the transmission of the gripper are also deflected, but in this case, this does not have to be taken into account as the deflection of the gripping force sensor is decoupled from the rest. The camera (Technisch Industrielles Industrie Mikroskop, Scientific Precision Instruments GmbH, Nieder-Olm, Germany) used records with a resolution of 768 × 576. When the recorded images are combined with the measured forces, the relationship between the deflection of the force sensors and the effective force can be read in the image. For this purpose, structures of a known length and the visible deflection of the force sensors are manually measured in the image. From this, the actual deflection can be calculated in µm.

The resulting force–deflection diagrams can be seen in Figure 9. The force sensors of gripper have sensitivities of 3.650 µm/mN on side 1 and 3.077 µm/mN on side 2. The sensitivities of the sensors of gripper B are 3.241 µm/mN on side 1 and 3.346 µm/mN on side 2. The offsets can also be determined.

#### 3.2.2. Sample Interaction

After calibration, an object—in this case, a generic rubber piece—is held between the gripper jaws (Figure 10). The object is intentionally slightly larger than the maximum unloaded distance between the gripper jaws to preload the force-sensing structures and show their reaction to the full scale of the pressure. The system pressure for actuator control is varied: from the measured maximum 0.12 MPa (minimum gripping force) to the measured minimum −0.06 MPa (maximum gripping force) and back. The process is repeated for each gripping force sensor on a gripper, and the microscope camera is always directed at the side to be examined. The air pressure is measured with a pressure sensor (DP-100, Panasonic, Kadoma, Osaka, Japan) along the supply line to the gripper. The actual air pressure in the actuator element is significantly lower as a constant airflow passes through the actuator when pressure is applied. This is due to the leakage of the actuator element, which is an inherent property of the design concept. The deflections of the gripping force sensors can be read out from the image. In order to help the user, specialized recognition structures are integrated into the gripping force sensors. They function like the vernier scale of a caliper: the opposing teeth serve as markers with defined distances and immediately allow a discrete determination of the sensor deflection on the basis of the microscope image without further aids. For comparison, the deflection is also determined by measuring the image as described during calibration. Examples of the values being read out in this way are shown for one of the sides of a gripper against the system pressure in Figure 11.

With the help of the previously determined sensitivities and offsets, the applied forces can be determined from the read deflections. It is shown that the two microgrippers can build up maximum force differences in a range of 3.0–5.7 mN in the case examined. All values are shown in Table 2.

The two reading procedures show certain differences with regard to the determined deflections. On average, the absolute differences between the methods are 0.20 mN (gripper A, side 1), −0.21 mN (gripper A, side 2), −1.23 mN (gripper B, side 1), and 0.41 mN (gripper B, side 2). The standard deviations are 0.82 mN (gripper B, side 2) or less. These values appear acceptable in light of the fact that the resolution of the recorded camera images shows approximately 4 µm per pixel which is more than 1 mN with the given sensitivites.

The readout structures divide the maximum deflection range of 100 µm into 20 steps of 5 µm length, each resulting in force differences of about 1.5 mN per step. This discretization can be seen clearly in the values of the reading structures. The values form steps in the graph. With finer structures, it would be possible to increase the resolution of the reading structures, provided that the resolution of the microscope camera is also increased. As SU-8 can be processed with a high aspect ratio, visually distinguishable test structures of 10 µm could be produced in the same process. Vernier scale readout structures in that size should be able to determine displacements of around 1 µm. The sensitivity can also be adjusted by a different dimensioning of the elastic force measurement springs. The existing spring elements are obviously designed for significantly higher forces than the actuator elements can provide. The low forces of the actuators can mainly be attributed to the design-related leakage, which prevents the full system pressure from building up in the actuator. This leakage is intrinsic to the actuation principle where two rigid planes (silicon and glass) have a compliant moving structure (SU-8) in between. The air pressure has to be trapped inside the actuator, but it can escape through the gap between the moving structures and the planes. Removing this gap would impede the actuator’s movement.

## 4. Conclusions

This paper introduced a new design for an SU-8-based microgripper, which is pneumatically driven and offers the possibility of image-based gripping force measurement. Perhaps most importantly, the gripper is realized by a very simple manufacturing process that allows low-cost batch production. This was achieved by shifting technical functions and, thus, manufacturing complexity into a reusable mounting. This turns the central gripper component into a disposable component that can be produced economically in bulk with just a single photolithography mask.

Simulated and experimental investigations were carried out on the force-measuring structures used in the microgripper. The good linearity of these sensors was confirmed, and the controllability of the design parameters was ascertained. Even if the gripping force of the gripper is quite low, the functionality of the present design could be demonstrated in practice. For illustration, Figure 12 shows the gripping procedure on a rubber test body. The application of novel image-based force measurement structures was also successfully demonstrated.

In future works, the actuator element of the gripper will be further improved to allow greater gripping forces and to avoid design-dictated leakage. It would also be useful to further improve the readout process of the force measurement structures.

## 5. Patents

Bütefisch, S.; Molnar, G.; Leester-Schädel, M.; Dietzel, A. Mikrogreifer und Verfahren zum Messen der Greifkraft eines Mikrogreifers. No. DE102017105463, 20 September 2018. 

## Figures and Tables

**Figure 1 micromachines-10-00707-f001:**
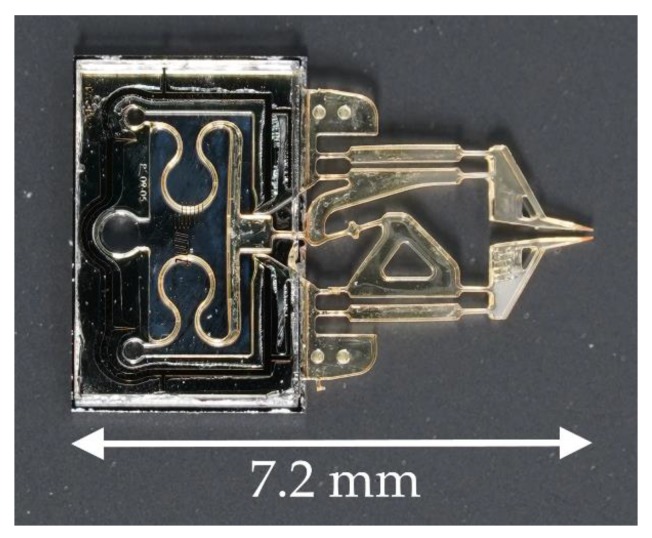
A pneumatic gripper design without force sensing capability as shown by Hoxhold [6], predecessor of the current iteration.

**Figure 2 micromachines-10-00707-f002:**
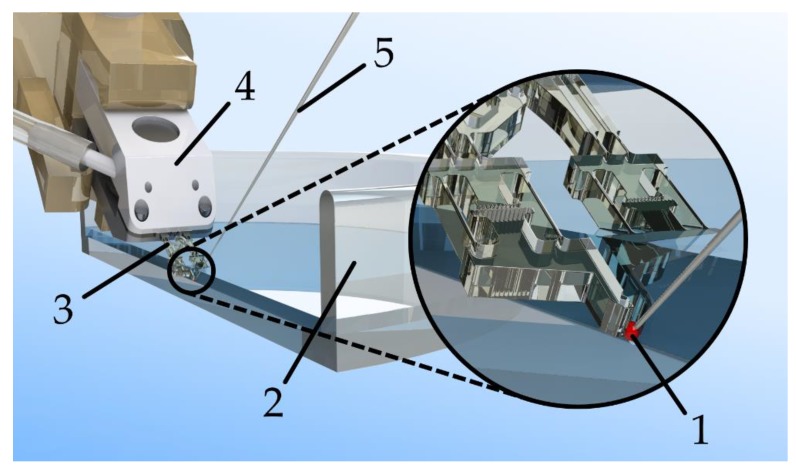
Concept sketch of the envisaged handling system allowing the patch clamp analysis of small cell clusters. A cell sample (**1**) is submerged in an electrolyte solution in a sample dish; (**2**) the sample is gripped with the aid of a disposable microgripper; (**3**) that is mounted on a reusable pneumatic actuator fixture; (**4**) the sample is contacted via a pipette; (**5**) filled with the same electrolyte. An electric current can thus be measured through the pipette, the sample, and the surrounding electrolyte.

**Figure 3 micromachines-10-00707-f003:**
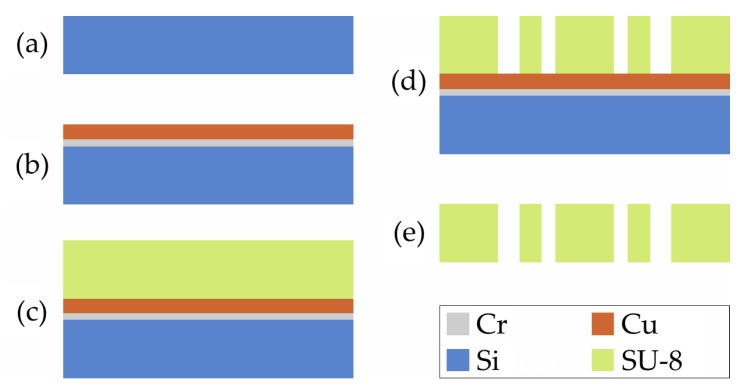
Schematic visualization of the manufacturing process for the disposable gripper component: (**a**). The silicon carrier wafer; (**b**). The sputtering of Cr and Cu; (**c**). Depositing the SU-8; (**d**). Structuring the SU-8; (**e**). The release by etching of the Cu sacrificial layer.

**Figure 4 micromachines-10-00707-f004:**
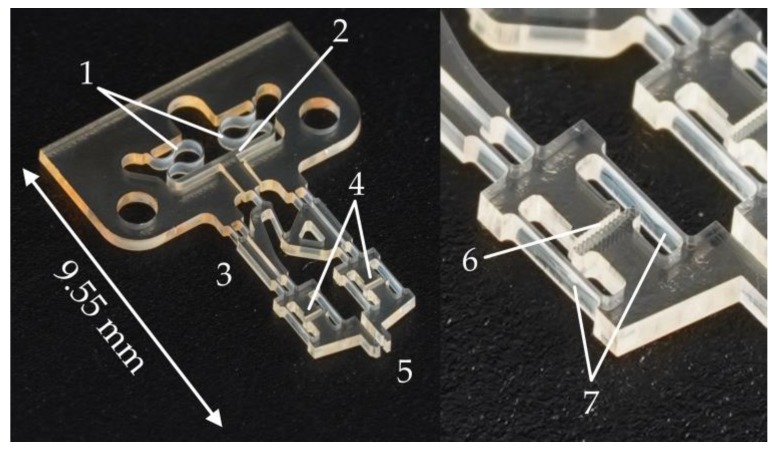
The disposable gripper component made of SU-8 with a larger partial view of the force measurement structure in one of its arms. (**1**): The flexible part of the pneumatic actuator. (**2**): The rigid pressure barrier of the pneumatic actuator. (**3**): The compliant hinge transmission. (**4**): Force sensors. (**5**): Gripping jaws. (**6**): The optical readout structure. (**7**): The force sensor beams.

**Figure 5 micromachines-10-00707-f005:**
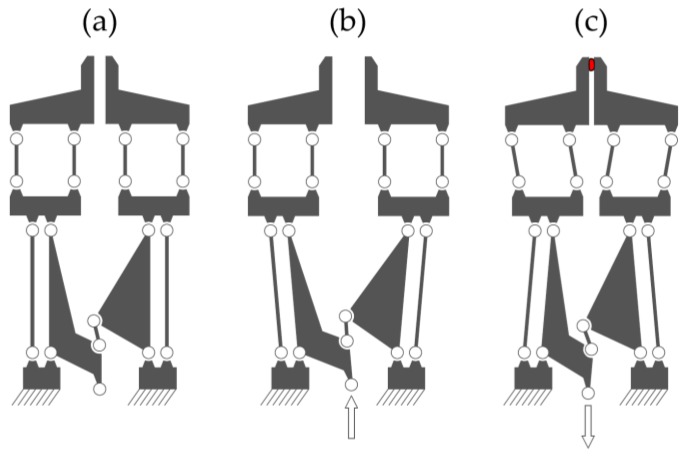
Simplified schematic view of the compliant hinge transmission. (**a**): No actuator force. (**b**): The actuator pushes (positive pressure). (**c**): The actuator pulls (negative pressure) and the gripping force sensors get deflected according to the force applied to the sample.

**Figure 6 micromachines-10-00707-f006:**
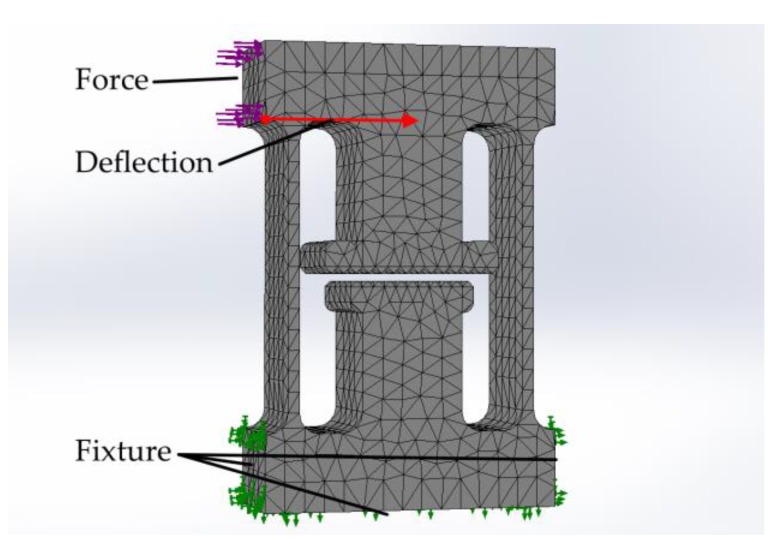
Model used for simulation.

**Figure 7 micromachines-10-00707-f007:**
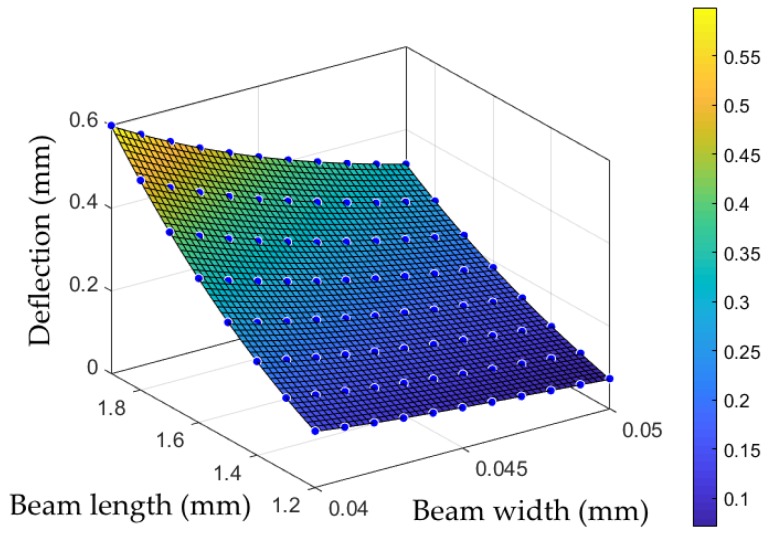
The simulation results of the sensor deflection (blue points) as a function of the geometric design parameters of spring beam length and width. The thickness of the SU-8 layer was set to 400 µm and the applied force to 30 mN. The colored area is numerically approximated to the simulation data by means of third-degree polynomials.

**Figure 8 micromachines-10-00707-f008:**
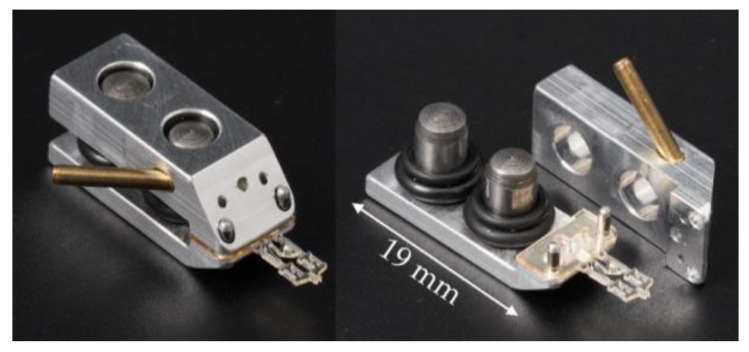
The central gripper component inserted into its mounting.

**Figure 9 micromachines-10-00707-f009:**
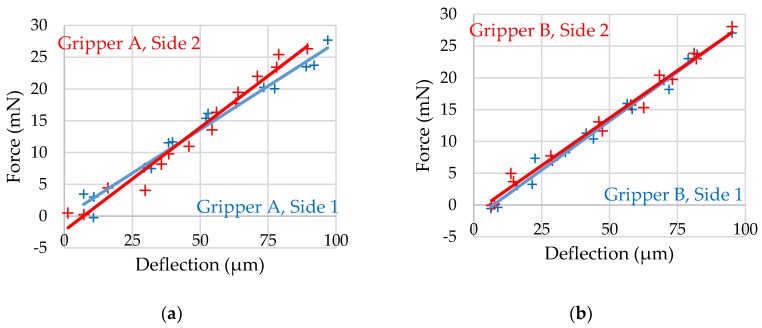
Force–deflection diagrams determined by image evaluation of the fully functional grippers for calibration of their force measurement structures: (**a**) Gripper A, side 1: offset: −0.104 mN, R²: 0.9740; side 2: offset: −0.297 mN, R²: 0.9697 (**b**) Gripper B, side 1: offset: −2.220 mN, R²: 0.9856; side 2: offset: −1.279 mN, R²: 0.9846.

**Figure 10 micromachines-10-00707-f010:**
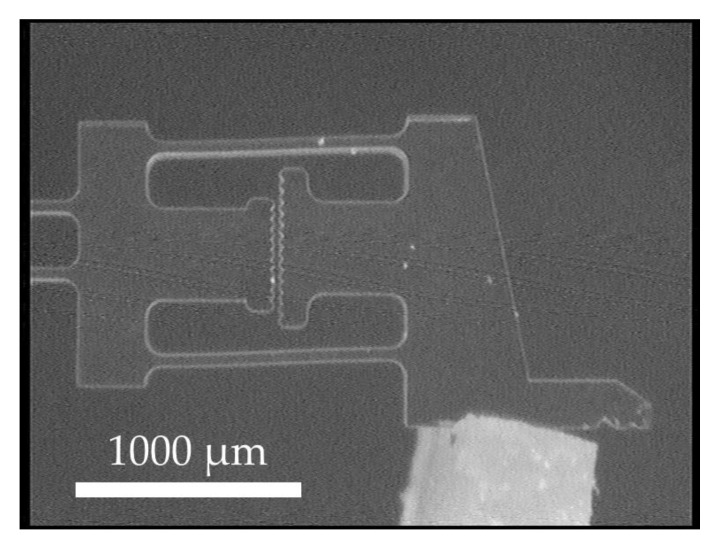
Gripping force sensor in an experiment with a gripped rubber sample. The deformation of the force-measuring structure and the displacement in the vernier readout structure are clearly visible.

**Figure 11 micromachines-10-00707-f011:**
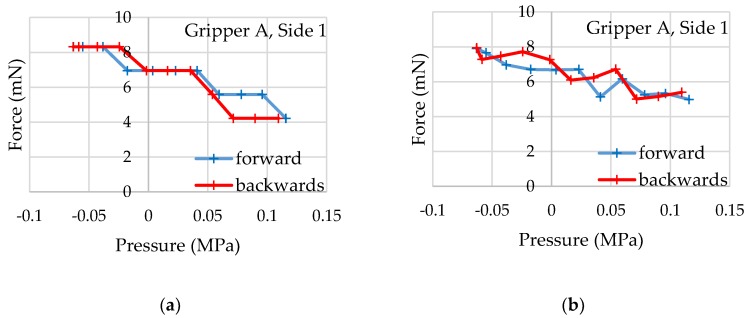
Forces determined by the elastic force measurement structures on gripper A, side 1, against the applied actuator pressure. (**a**) Values are read on the vernier scale; (**b**) Values are determined by measuring the image.

**Figure 12 micromachines-10-00707-f012:**
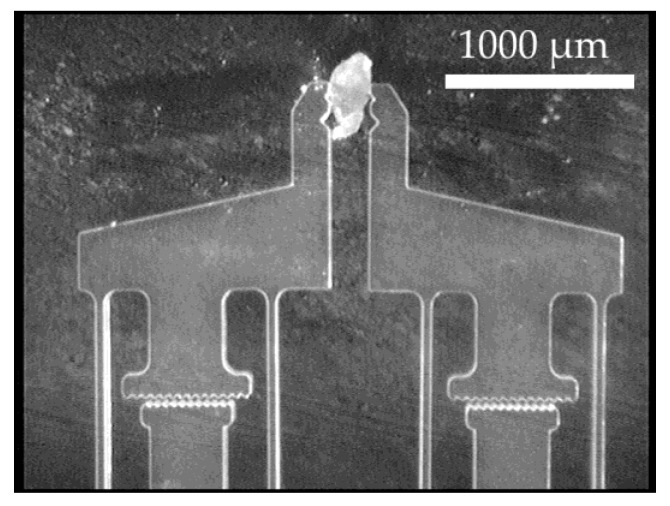
A demonstration of the gripping process on a rubber test body with the newly designed microgripper.

**Table 1 micromachines-10-00707-t001:** Sensitivities, offsets, and coefficients of determination (R²) of the force-measuring structures on nine dummy gripper samples of nominally identical shape.

Sample No.	Sensitivity (µm/mN)	Offset (mN)	R²
1	3.415	−0.736	0.9946
2	3.399	−0.663	0.9960
3	3.348	−0.063	0.9951
4	3.428	−0.601	0.9948
5	3.361	−0.896	0.9940
6	3.310	−1.187	0.9933
7	3.182	−0.535	0.9956
8	3.382	−0.949	0.9932
9	3.211	−0.638	0.9960

**Table 2 micromachines-10-00707-t002:** The difference between the largest and the smallest effective gripping forces. ΔForce 1 is determined via the vernier structures. ΔForce 2 is determined by manually measuring the deflection in the image of the force sensors.

Gripper, Side	ΔForce 1 (mN)	ΔForce 2 (mN)
A, 1	4.1	3.0
A, 2	3.3	5.1
B, 1	4.6	5.1
B, 2	4.5	5.7

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
