# Peer review of "A Disposable Pneumatic Microgripper for Cell Manipulation with Image-Based Force Sensing"

_micromachines, 2019, doi:10.3390/mi10100707_

Round 1
Reviewer 1 Report
Please see attached

Author Response
Response to Reviewer 1
Point 1: The introduction must be rewritten presenting the state of the art more accurately. Many works of various research groups are ignored and should be considered. In fact, literature about microgrippers is very vast [1] and the presented investigation should be set in the current scenario. For example, Femtotools refers to the work of Beyeler [2], that should be considered as reference instead of the website. Pneumatic actuation should be motivated comparing it with other actuation strategies at the microscale in a more comprehensive way [3-5].
Response 1: Additional references have been added to the introduction. A microscale comparison between pneumatic actuation and other actuation principles would certainly be interesting but seems to exceed the scope of this manuscript as all common actuation principles can be ruled out for the presented purpose based on the fact alone that they are electrically driven (see Response 2).
Point 2: Many statements needs to be motivated in a deeper manner. For example, in the phrase "Electromagnetic fields can exert disturbing influences on very weak electrophysiological signals from cells and organoids", there are no data about the magnitude of these fields and the signals, nor the references. The same holds for "This could be disadvantageous for a gripped cell cluster, as it could be forced out of the tilted gripper jaw" regarding the rotational movement, because it depends on the position of the center of rotations [6].
Response 2: The influences that electromagnetic systems near a sample have on patch clamp measurements are currently unknown. “Electromagnetic fields can exert disturbing influences […]“ has been changed to „Electromagnetic fields could exert disturbing influences […]“ to reflect this. That is why the approach presented in this manuscript circumvents any of these potential influences entirely by focusing on non-electric concepts and therefore reducing the potential risk of the design being unfit for its intended purpose. In a similar manner, grippers with jaws that tilt are considered but not used for the design. This way, potential problems are circumvented before they can arise.
Point 3: The paragraphs regarding the motivations of the fabrication (lines 50-58) and the sensor (59-66) are also weak: they are not clear and should be better explained, maybe with some pictures.
Response 3: The description of the predecessor gripper design, its actuation mechanism and its fabrication have been reworked. This should make it clearer why a redesign was necessary to enable batch processing. The text has also been reworked to better show why a new approach for measuring the gripping forces is examined.
Point 4: In section 2.1, it should be explained why two layers (Cr and Cu) are used for "better adhesion".
Response 4: The text could indeed be misinterpreted and is now improved. The Cr-layer enhances the adhesion of the Cu-layer. The sole purpose of the copper is to serve as a sacrificial layer.
Point 5: Considering the design, in section 2.2, a schematic representation of the system kinematics could help the reader regarding the working principle and the "quasi-parallel displacement" obtained with the flexure-based parallelogram linkage. Also many details about the FEA simulations are missing, such as mesh, constraints, material,...
Response 5: A schematic view has been added. It shows the microgripper’s transmission in its three states neutral, positive pressure and negative pressure to help the reader understand the kinematics.
Additional information about the simulation setup has been added including a figure of the used model.
Point 6: The experimental setup in section 3.1 is not clear, with the device, the commercial sensor and the linear drive; a schematic could be useful.
Response 6: The text has been reworked to make the experimental setup clearer.
Point 7: In section 3.2, relative difference should be considered regarding figure 9. Why forces were not determined also by means of FEA simulations?
Response 7: Relative differences hold the same amount of information as the absolute differences which are already mentioned. However, the latter are easier set in perspective with the limited resolution of the recordings.
Simulations of the gripping force sensors are described in section 2.2. The transmission itself is both described in detail and simulated in [7]. For the experiments in section 3.2 the gripper’s actuator and transmission were treated as a black box as it is currently not possible to determine the pneumatic pressure inside the actuator element due to the design-inherent leakage. This makes use of the fact that the gripping force sensors’ deflections are decoupled from the compliant transmission thanks to their integration into the gripping jaws.
[1] A. Dochshanov, M. Verotti, and N.P. Belfiore. A comprehensive survey on microgrippers design: Operational strategy. Journal of Mechanical Design, Transactions of the ASME, 139(7), 2017.
[2] Felix Beyeler, Adrian Neild, Stefano Oberti, Dominik J Bell, Yu Sun, Jrg Dual, and Bradley J Nelson. Monolithically fabricated microgripper with integrated force sensor for manipulating microobjects and biological cells aligned in an ultrasonic field. Journal of microelectromechanical systems, 16(1):7-15, 2007.
[3] Andrew J. Fleming. A review of nanometer resolution position sensors: Operation and performance. Sensors and Actuators A: Physical, 190:106 - 126, 2013.
[4] D J Bell, T J Lu, N A Fleck, and S M Spearing. MEMS actuators and sensors: observations on their performance and selection for purpose. Journal of Micromechanics and Microengineering, 15(7):S153, 2005.
[5] Ernst Thielicke and Ernst Obermeier. Microactuators and their technologies. Mechatronics, 10:431 - 455, 2000.
[6] M Verotti. Analysis of the center of rotation in primitive flexures: Uniform cantilever beams with constant curvature. Mechanism and Machine Theory, 97:29-50, 2016.
[7] Hoxhold, B. Mikrogreifer und aktive Mikromontagehilfsmittel mit integrierten Antrieben. Dissertation, 2010, ISBN 9783832296780.
Reviewer 2 Report
This paper presents a new design of pneumatic microgripper with optically readable force sensors. This new design is very interesting and has potentials in many important applications such as patch clamping. Considering the scientific merit, the paper may be accepted if the following issues are addressed:
In the abstract, the author wrote: “An optically readable force sensor suitable for a feedback loop has been integrated in order to enable gripping with a controlled force.” This well explained the advantage of this new design. But it seems the author didn’t achieve closed-loop control for the gripping force. Please clarify. Experiments reported in Table two should be repeated for several times to add the standard deviation to prove the consistency of the measurement. The authors need to explain how they measured the displacement and use it for calculating the forces. Is the displacement in the image detected manually, or is there an automated image processing algorithm is used? This must be clarified so other researchers can repeat this technology. In the last paragraph of Section 3, the author mentioned some limitations by saying “The low forces of the actuators can mainly be attributed to the design-related leakage, which prevents the full system pressure from building up in the actuator.” Is there a way to address this problem? Or is it a fundamental drawback of this design. Please clarify.Author Response
Response to Reviewer 2
Point 1: In the abstract, the author wrote: “An optically readable force sensor suitable for a feedback loop has been integrated in order to enable gripping with a controlled force.” This well explained the advantage of this new design. But it seems the author didn’t achieve closed-loop control for the gripping force. Please clarify.
Response 1: A feedback loop with automated readout in mind is planned but not yet implemented. The gripping force sensors are designed with this in mind. That is why the text mentions them to be “suitable for a feedback loop”. The text has been rephrased to avoid misinterpretation.
Point 2: Experiments reported in Table two should be repeated for several times to add the standard deviation to prove the consistency of the measurement.
Response 2: The beginning of section 3.2 has been updated to clarify that the experiments with test bodies are meant as examples to demonstrate the force sensor’s usage in practical application. The mechanical characterization with a larger set of samples is subject of section 3.1.
Point 3: The authors need to explain how they measured the displacement and use it for calculating the forces. Is the displacement in the image detected manually, or is there an automated image processing algorithm is used? This must be clarified so other researchers can repeat this technology.
Response 3: In section 3.2.1 Force Sensor Calibration it is described how the displacements are gathered from the images. This happens manually which is now specifically mentioned to avoid misunderstandings. Section 3.2.2 mentions both ways of gathering the deflections from the images: by reading the vernier scale and by measuring the image as described in section 3.2.1. It is also mentioned in 3.2.2 that the previously (during calibration) gathered values for sensitivity and offset are used to calculate the equivalent forces.
Point 4: In the last paragraph of Section 3, the author mentioned some limitations by saying “The low forces of the actuators can mainly be attributed to the design-related leakage, which prevents the full system pressure from building up in the actuator.” Is there a way to address this problem? Or is it a fundamental drawback of this design. Please clarify.
Response 4: It has been added to the text that a gap that allows air to pass (leakage) is intrinsic to this actuation concept. Without the gap the actuator could not move.
Round 2
Reviewer 1 Report
The paper has been revised according to the comments
Reviewer 2 Report
All my previous comments are well addressed.